# 4D Panoptic Scene Graph Generation

Jingkang Yang[1], Jun Cen[2], Wenxuan Peng[1], Shuai Liu[3], Fangzhou Hong[1],
Xiangtai Li[1], Kaiyang Zhou[4], Qifeng Chen[2], Ziwei Liu[1] ✉

[1]S-Lab, Nanyang Technological University
[2]The Hong Kong University of Science and Technology
[3]Beijing University of Posts and Telecommunications    [4]Hong Kong Baptist University

`https://github.com/Jingkang50/PSG4D`

**(a) Visual Input from the 4D Dynamic World**

**(c) Reasoning & Planning**

**(b) PSG-4D: 4D Panoptic Scene Graph**

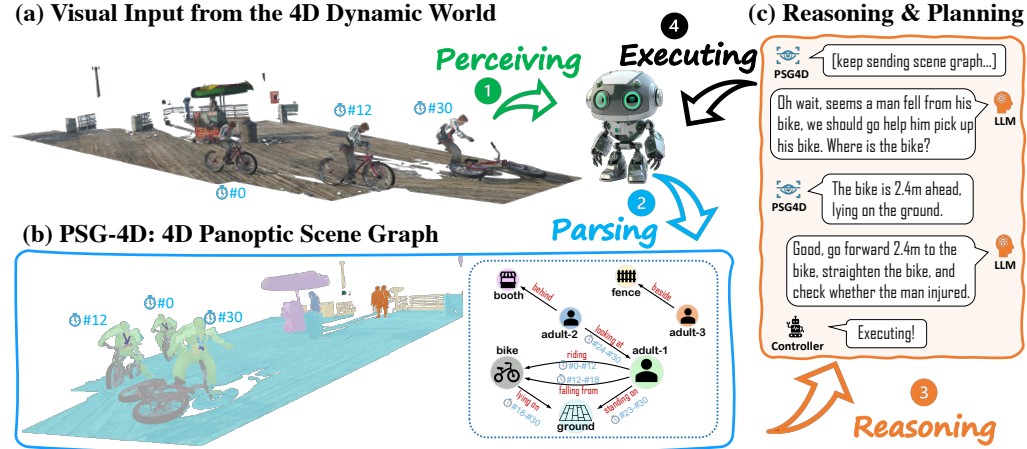

Figure 1: **Conceptual illustration of PSG-4D.** PSG-4D is essentially a spatiotemporal representation capturing not only fine-grained semantics in image pixels (i.e., panoptic segmentation masks) but also the temporal relational information (i.e., scene graphs). In (a) and (b), the model abstracts information streaming in RGB-D videos into (i) nodes that represent entities with accurate location and status information and (ii) edges that encapsulate the temporal relations. Such a rich 4D representation serves as a bridge between the PSG-4D system and a large language model, which greatly facilitates the decision-making process, as illustrated in (c).

## Abstract

We are living in a three-dimensional space while moving forward through a fourth dimension: time. To allow artificial intelligence to develop a comprehensive understanding of such a 4D environment, we introduce *4D Panoptic Scene Graph (PSG-4D)*, a new representation that bridges the raw visual data perceived in a dynamic 4D world and high-level visual understanding. Specifically, PSG-4D abstracts rich 4D sensory data into nodes, which represent entities with precise location and status information, and edges, which capture the temporal relations. To facilitate research in this new area, we build a richly annotated PSG-4D dataset consisting of 3K RGB-D videos with a total of 1M frames, each of which is labeled with 4D panoptic segmentation masks as well as fine-grained, dynamic scene graphs. To solve PSG-4D, we propose PSG4DFormer, a Transformer-based model that can predict panoptic segmentation masks, track masks along the time axis, and generate the corresponding scene graphs via a relation component. Extensive experiments on the new dataset show that our method can serve as a strong baseline for future research on PSG-4D. In the end, we provide a real-world application example to demonstrate how we can achieve dynamic scene understanding by integrating a large language model into our PSG-4D system.

---

✉ Corresponding author. Contact: {`jingkang001`, `ziwei.liu`}`@ntu.edu.sg`

37th Conference on Neural Information Processing Systems (NeurIPS 2023).

# 1 Introduction

The emergence of intelligent agents, autonomous systems, and robots demands a profound understanding of real-world environments [1, 2, 3, 4, 5, 6]. This understanding involves more than just recognizing individual objects – it requires an intricate understanding of the relationships between these objects. In this context, research on Scene Graph Generation (SGG) [7], has sought to provide a more detailed, relational perspective on scene understanding. In this approach, scene graphs represent objects as nodes and their relationships as edges, offering a more comprehensive and structured understanding of the scene [8, 7, 9, 10, 11]. Panoptic Scene Graph Generation (PSG) [12] expands the scope of SGG to encompass pixel-level precise object localization and comprehensive scene understanding, including background elements. Then PSG has been further extended to the domain of videos [13] with the inspiration from Video Scene Graph Generation (VidSGG) [14, 15].

The utility of scene graphs also extends into the realm of 3D perception, introducing the concept of 3D Scene Graphs (3DSG) [16, 17]. 3DSGs offer a precise representation of object locations and inter-object relationships within three-dimensional scenes [18, 19]. Despite these developments, the existing approaches have not fully integrated dynamic, spatio-temporal relationships, particularly those involving human-object and human-human interactions. Consider Figure 1 as an illustrative example. Traditional 3D scene graph methods may recognize the static elements of this scene, such as identifying a booth situated on the ground. However, a more ideal, advanced, and dynamic perception is required for real-world scenarios. For instance, a system should be capable of identifying a dynamic event like a person who has fallen off their bike, so that it could then comprehend the necessity to offer assistance, like helping the person stand up and stabilize their bike.

Therefore, our work takes a significant step towards a more comprehensive approach to sensing and understanding the world. We introduce a new task, the 4D Panoptic Scene Graph (PSG-4D), aiming to bridge the gap between raw visual inputs in a dynamic 4D world and high-level visual understanding. PSG-4D comprises two main elements: nodes, representing entities with accurate location and status information, and edges, denoting temporal relations. This task encapsulates both spatial and temporal dimensions, bringing us closer to a true understanding of the dynamic world.

To facilitate research on this new task, we contribute an extensively annotated PSG-4D dataset that is composed of 2 sub-sets, PSG4D-GTA and PSG4D-HOI. The PSG4D-GTA subset consists of 67 RGB-D videos with a total of 28K frames, selected from the SAIL-VOS 3D dataset [20] collected from the video game Grand Theft Auto V (GTA-V) [21]. The PSG4D-HOI subset is a collection of 3K egocentric real-world videos sampled from the HOI4D dataset [22]. All frames in either of the subset are labeled with 4D panoptic segmentation masks as well as fine-grained, dynamic scene graphs. We believe this dataset will serve as a valuable resource for researchers in the field.

To tackle this novel task, we propose a unified framework called PSG4DFormer. This unified structure encapsulates two primary components: a 4D Panoptic Segmentation model and a Relation model. The 4D Panoptic Segmentation model is designed to accommodate both RGB-D and point cloud data inputs, yielding a 4D panoptic segmentation. This output comprises 3D object masks, which are continuously tracked across temporal dimensions. Then, the Relation model accepts these 3D mask tubes and utilizes a spatial-temporal transformer architecture to delineate long-term dependencies and intricate inter-entity relationships, subsequently yielding a relational scene graph. Through extensive experiments, we demonstrate the effectiveness of the proposed PSG-4D task and the PSG4DFormer model. Our work constitutes a pivotal step towards a comprehensive understanding of dynamic environments, setting the stage for future research in this exciting and crucial area of study.

In summary, we make the following contributions to the community:

- **A New Task**: We propose a novel scene graph generation task focusing on the prediction of 4D panoptic scene graphs from RGB-D or point cloud video sequences.

- **A New Dataset**: We provide a PSG-4D dataset, which covers diverse viewpoints: (i) a third-view synthetic subset (PSG4D-GTA) and (ii) an egocentric real-world subset (PSG4D-HOI).

- **A Unified Framework**: We propose a unified two-stage model composed of a feature extractor and a relation learner. In addition, we offer demo support for both synthetic and real-world scenarios to facilitate future research and real-world applications.

- **Open-Source Codebase**: We open-source our codebase to facilitate future PSG-4D research.

## 2  Related Work

**Scene Graph Generation (SGG)**    SGG transforms an image into a graph, where nodes represent objects and edges represent relationships [7]. Several datasets [23] and methods, including two-stage [8, 7, 9, 10, 11] and one-stage models [24, 12, 25], have been developed for SGG. Video scene graph generation (VidSGG) extends SGG to videos with notable datasets [14, 15, 26]. Despite progress, limitations remain in SGG and VidSGG due to noisy grounding annotations caused by coarse bounding box annotations and trivial relation definitions. Recent work on panoptic scene graph generation (PSG) [12, 27, 28, 29, 30, 31] has attempted to overcome these issues, and PVSG [13, 32] further extends it into the video domain. This paper presents an extension of PSG into a 4D dynamic world, meeting the needs of active agents for precise location and comprehensive scene understanding.

**3D Scene Graph Generation**    3D Scene Graphs (3DSGs) offer a precise 3D representation of object locations and inter-object relationships, making them a vital tool for intelligent agents operating in real-world environments [16, 17]. 3DSGs can be categorized into flat and hierarchical structures [33]. The former represents objects and relationships as a simple graph [18, 19], while the latter layers the structures of 3D scenes [34, 35]. Recent 3DSG techniques [19] employ PointNet [36] with 3D object detectors on point clouds or RGBD scans, generating 3D graphs via graph neural networks [18]. Some settings, such as Kimera [37], emphasize pairwise spatiotemporal status to facilitate task planning, while incremental 3DSG necessitates agents to progressively explore environments [38]. However, these graphs largely represent positional relations, lacking dynamic spatiotemporal relations like human-object interactions and human-human relations.

**4D Perception**    Research on 4D perception can be divided by the specific data format they use. The first one is RGB-D video, which can be easily obtained using cheap sensors, *e.g.* Kinect, and iPhone. With the additional depth data, more geometric and spatial information can be used for reliable and robust detection [39, 40, 41] and segmentation [42, 43, 44, 45]. For RGB-D video, the depth input is usually treated like images. But for point clouds video, 3D or higher dimension convolutions [46, 47, 48, 49] are more commonly used, especially on LiDAR point cloud videos for autonomous driving perception system. In this work, beyond the 4D panoptic segmentation, we focus on more daily scenes and pursue a more high-level and structured understanding of 4D scenes by building 4D scene graphs.

## 3  The PSG-4D Problem

The PSG-4D task is aimed at generating a dynamic scene graph, which describes a given 4D environment. In this context, each node corresponds to an object, while each edge represents a spatial-temporal relation. The PSG-4D model ingests either an RGB-D video sequence or a point cloud video sequence, subsequently outputting a PSG-4D scene graph $\mathbf{G}$. This graph is composed of 4D object binary mask tubes $\mathbf{M}$, object labels $\mathbf{O}$, and relations $\mathbf{R}$.

The object binary mask tubes, $\mathbf{m}_i \in \{0,1\}^{T \times H \times W \times 4}$, express the 3D location and extent of the tracked object $i$ over time ($T$) in the case of an RGB-D sequence input, while $\mathbf{m}_i \in \{0,1\}^{T \times M \times 6}$ is used for point cloud video inputs. Here, 4 denotes RGB-D values, and 6 represents XYZ plus RGB values. M stands for the number of point clouds of interest. The object label, $o_i \in \mathbb{C}^O$, designates the category of the object. The relation $r_i \in \mathbb{C}^R$ represents a subject and an object linked by a predicate class and a time period. $\mathbb{C}^O$ and $\mathbb{C}^R$ refer to the object and predicate classes, respectively. The PSG-4D task can be mathematically formulated as:

$$\Pr(\mathbf{G} \mid \mathbf{I}) = \Pr(\mathbf{M}, \mathbf{O}, \mathbf{R} \mid \mathbf{I}), \qquad (1)$$

where $\mathbf{I}$ represents the input RGB-D video sequence or point cloud representation.

**Evaluation Metrics**    For evaluating the performance of the PSG-4D model, we employ the R@K and mR@K metrics, traditionally used in the scene graph generation tasks. R@K calculates the triplet recall, while mR@K computes the mean recall, both considering the top K triplets from the PSG-4D model. A successful recall of a ground-truth triplet must meet the following criteria: 1) correct category labels for the subject, object, and predicate; 2) a volume Intersection over Union (vIOU) greater than 0.5 between the predicted mask tubes and the ground-truth tubes. When these criteria are satisfied, a soft recall score is recorded, representing the time vIOU between the predicted and the ground-truth time periods.

Table 1: **Illustration of the PSG-4D dataset and related datasets.** Unlike the static 3D indoor scenes usually found in 3DSG datasets, the PSG-4D dataset introduces dynamic 3D videos, each annotated with panoptic segmentation. Various 3D video datasets were evaluated as potential sources for PSG-4D, resulting in the creation of two subsets: PSG4D-GTA and PSG4D-HOI. Regarding annotations, PS represents Panoptic Segmentation, BB represents Bounding Box, SS represents Semantic Segmentation, KP represents key points, and PC represents point clouds. TPV represents third-person-view.

| Dataset | Type | Scale | View | #ObjCls | #RelCls | Annotation | Year |
|---|---|---|---|---|---|---|---|
| 3DSSG [18] | 3DSG | 363K RGB-D images, 1482 scans, 478 scenes, | TPV | 534 | 40 | 3D model, 3D graph | 2020 |
| Rel3D [50] | 3DSG | 27K RGB-D images, 9990 3D Scenes | TPV | 67 | 30 | 3D model | 2020 |
| ScanNet [51] | 3D Images | 2.5M RGB-D images, 1513 indoor scenes | TPV | 20 | - | SS, 3D model | 2017 |
| Matterport 3D [52] | 3D Images | 194,400 RGB-D images, 90 building-scale scenes | TPV | 40 | - | SS, 3D model | 2017 |
| Nuscenes [53] | 2D Video+PC | 1K videos (avg. 20s), 1.3M pointclouds | Vehicle | 23 | - | 3D BB | 2020 |
| WAYMO [54] | 2D Video+PC | 1.2K videos (avg. 20s), 177K pointclouds | Vehicle | 20 | - | 2D BB, 3D BB | 2020 |
| Sail-VOS 3D [55] | 3D Video | 484 videos, 238K RGB-D image, 6807 clips | egocentric | 178 | - | SS, 3D model | 2021 |
| HOI4D [56] | 3D Video | 4K videos, 2.4M RGB-D image, 610 indoor scenes | egocentric | 16 | 11 | PS, KP | 2022 |
| EgoBody [57] | 3D Video | 125 videos, 199K RGB-D images, 15 indoor scenes | egocentric, TPV | 36 | 13 | 3D model, KP | 2022 |
| **PSG4D-GTA** | PSG4D | 67 videos (avg. 84s), 28K RGB-D images, 28.3B pointclouds | TPV | 35 | 43 | PS, 4DSG | 2023 |
| **PSG4D-HOI** | PSG4D | 2973 videos (avg. 20s), 891K RGB-D images, 282 indoor scenes | egocentric | 46 | 15 | PS, 4DSG | 2023 |

# 4 The PSG-4D Dataset

This section outlines the development of the PSG-4D dataset. We begin by exploring existing datasets that inspired the creation of PSG-4D, followed by a presentation of its statistics, and finally a brief overview of the steps involved in its construction.

## 4.1 Leveraging Existing Datasets for PSG-4D

Rather than constructing the PSG-4D dataset from the ground up, we sought to evaluate whether currently available datasets could either directly support or be adapted for the PSG-4D task. As shown in Table 1, our initial exploration focused on 3D datasets, including 3D scene graph datasets like 3DSGG [18] and Rel3D [50], along with more conventional 3D datasets such as ScanNet [51] and Matterport 3D [52]. However, while these datasets can be used to reconstruct entire scenes and can generate 3D videos accordingly, the resulting scenes remain static and lack dynamic elements.

We then shifted our focus to video datasets containing 3D information. Autonomous driving datasets such as Nuscenes [53] and WAYMO [54] incorporate point cloud videos, particularly bird's-eye view footage. Nevertheless, the vehicles within these scenes are only captured in 2D video. While this technically constitutes a dynamic 4D scene, it does not align well with the objectives of this study. The dynamic relations in traffic scenarios are relatively limited, and our goal is to develop a visual understanding model for embodied AI [58, 59, 60, 61] that captures 3D scenes from the agent's perspective, not a bird's-eye view.

Another category of 3D videos uses RGB-D sequences as input, which can be easily converted into point clouds. This data format aligns perfectly with the operation of intelligent agents, mimicking human perception, which captures continuous RGB images with depth. Thankfully, recent datasets like SAIL-VOS 3D [55], HOI4D [56], and EgoBody [57] have adopted this approach. While SAIL-VOS 3D uses synthetic data from the GTA game [21], the HOI4D dataset captures egocentric RGB-D videos of simple tasks, such as tool picking. On the other hand, the EgoBody dataset [57] records office activities like conversations, but lacks segmentation annotation and is primarily intended for human pose reconstruction. Despite its wealth of videos, the object interaction in EgoBody is limited. In the medical domain, 4D-OR [60] excels in providing detailed depictions of surgical scenes, showcasing its specialized utility. To cater to a broader spectrum of research applications, we formulated the PSG-4D dataset, integrating the versatile strengths of the SAIL-VOS 3D [55] and HOI4D [56] datasets.

## 4.2 Dataset Statistics

Figure 2 presents a selection of four video frames, drawn from both the PSG4D-GTA and PSG4D-HOI datasets. Each frame is an RGB-D video with corresponding panoptic segmentation annotations. Underneath each scene, we depict the associated scene graph and statistical word clouds. Annotators constructed these scene graphs as triplets, complete with frame duration. The PSG4D-GTA dataset is

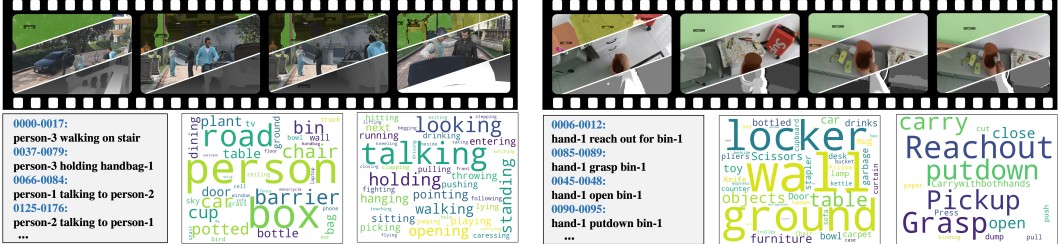

(a) **PSG4D-GTA** (Synthetic, Third-Person View)  (b) **PSG4D-HOI** (Real-World, Egocentric)

Figure 2: **The Examples and Word Clouds of PSG-4D dataset.** The PSG-4D dataset contains 2 subsets, including (a) PSG4D-GTA selected from the SAIL-VOS 3D [20] dataset, and (b) PSG4D-HOI from HOI4D [22] dataset. We selected 4 frames of an example video from each subset. Each frame has aligned RGB and depth with panoptic segmentation annotation. The scene graph is annotated in the form of triplets. The word cloud for object and relation categories in each dataset is also represented.

particularly noteworthy for its composition: it contains 67 videos with an average length of 84 seconds, amounting to 27,700 RGB-D images, 28.3 billion point clouds, and comprises 35 object categories, and 43 relationship categories. This synthetic dataset was captured from a third-person perspective. In contrast, the PSG4D-HOI dataset is compiled from an egocentric perspective, providing a different context for analysis. It includes 2,973 videos with an average duration of 20 seconds, equating to 891,000 RGB-D images across 282 indoor scenes. This dataset includes 46 object categories and 15 object-object relationship categories, offering a diverse range of real-world data for the study. The combination of these two datasets offers a comprehensive understanding of 4D environments due to their complementary nature. A statistical overview of both datasets can be found in the final two rows of Table 1.

### 4.3   Dataset Construction Pipeline

As outlined in Section 4.1, the PSG4D-GTA is built upon the SAIL-VOS 3D dataset, while the PSG4D-HOI is derived from the HOI4D dataset. To adapt the SAIL-VOS 3D dataset for our purpose, we commenced with a comprehensive review of all 178 GTA videos within the dataset. This stage involved a meticulous elimination process to exclude videos containing NSFW content, resulting in a refined pool of 67 videos. The SAIL-VOS 3D dataset, which is equipped with 3D instance segmentation, required additional annotation for background elements to integrate panoptic segmentation. Leveraging the PVSG annotation pipeline, we employed an event detection method [62] to isolate the key frames. The background elements within these key frames were subsequently annotated using the pre-annotation provided by the SAM model [63]. Upon completion of key frame annotations, the AOT method [64] was utilized to propagate the segmentation across the entire video sequence. The final step involved overlaying the instance segmentation on the stuff segmentation, thereby completing the process. The HOI-4D dataset, devoid of NSFW content, already provides a 4D panoptic segmentation. Consequently, we included all videos from the HOI-4D dataset in the PSG4D-HOI dataset without further modifications.

Upon completion of 4D panoptic segmentation annotation, we proceed to annotate the dynamic scene graph according to the masks. Although HOI4D includes action annotation concerning the person, it doesn't account for interactions between objects. Nevertheless, certain actions such as "pick up" are appropriately considered predicates, and we automatically position the key object in the video to form a subject-verb-object triplet. Once the automatic-annotated dataset is prepared, we ask annotators to review and revise the pre-annotations to ensure accuracy. As SAIL-VOS 3D lacks all kinds of relational annotation, we commence scene graph annotation from scratch. The entire annotation process is diligently executed by the authors around the clock.

## 5   Methodology

This section details a unified pipeline PSG4DFormer for addressing the PSG-4D problem. As shown in Figure 3, our approach comprises two stages. The initial 4D panoptic segmentation stage aims to segment all 4D entities, including objects and background elements in Figure 3 (a), with the accurate

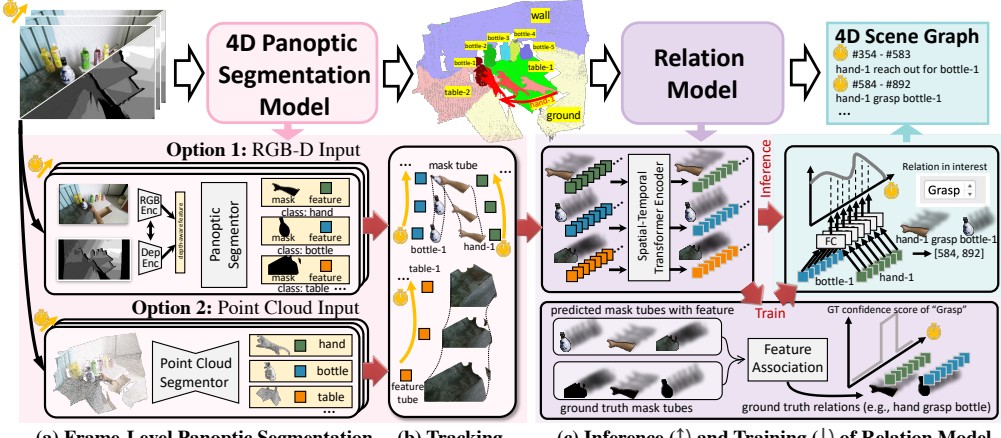

(a) Frame-Level Panoptic Segmentation   (b) Tracking   (c) Inference (↑) and Training (↓) of Relation Model

Figure 3: **Illustration of the PSG4DFormer pipeline.** This unified pipeline supports both RGB-D and point cloud video inputs and is composed of two main components: 4D panoptic segmentation modeling and relation modeling. The first stage seeks to obtain the 4D panoptic segmentation mask for each object, along with its corresponding feature tube spanning the video length. This is accomplished with the aid of (a) frame-level panoptic segmentation and (b) a tracking model. The subsequent stage (c) employs a spatial-temporal transformer to predict pairwise relations based on all feature tubes derived from the first stage.

temporal association in Figure 3 (b). We extract features for each object and obtain feature tubes according to tracking results for subsequent relation modeling in Figure 3 (c).

## 5.1   4D Panoptic Segmentation Modeling

As specified in Section 3, given a 3D video clip input, such as an RGB-D sequence of $\mathbf{I} \in \mathbb{R}^{T \times H \times W \times 4}$ or a point cloud sequence of $\mathbf{I} \in \mathbb{R}^{T \times M \times 6}$, the initial stage's goal is to segment and track each pixel non-overlappingly. The model predicts a set of video clips with the output of $(\mathbf{m}_i, \mathbf{q}_i, p_i)_{i=1}^N$, where $\mathbf{m}_i$ denotes the tracked object mask tube, $\mathbf{q}_i$ denotes the tracked feature tube, and $p_i$ represents the probability of the object belonging to each category. $N$ is the number of entities, encompassing things and stuff classes.

**Frame-Level Panoptic Segmentation with RGB-D Sequence**     Given the dual input of RGB and depth images, we adopt a separation-and-aggregation gate (SA-Gate) [65] to efficiently blend information from both modalities. This combined feature set, enriched with data from both inputs, is then fed into a robust Mask2Former [4] for frame-level panoptic segmentation. In the inference stage, at the frame $t$, given an RGB-D image $\mathbf{I}$, the Mask2Former with SA-Gate directly outputs a set of object query features $q_i^t \in \mathbb{R}^d, i = 1, \ldots, N$, each $q_i^t$ representing one entity at the frame $t$.

**Frame-Level Panoptic Segmentation with Point Cloud Sequence**     Apart from perceiving point cloud sequences directly, 3D point cloud coordinates can be calculated and converted from RGB-D data. This conversion involves computing the Normalized Device Coordinates (NDC) using the depth map and projecting the NDC to world coordinates using the transformation matrix provided. We retain only points with a depth below a defined threshold $\lambda$, discarding distant, less relevant elements like far-off mountains. To leverage texture information from the image, point cloud coordinates can be augmented with corresponding RGB values, creating a colorful point cloud representation $\mathbf{P} \in \mathbb{R}^{M \times 6}$, where $M$ is the total number of points in a frame.

We employ DKNet [66], a state-of-the-art indoor segmentation method, as our point cloud segmentation network. It processes input point clouds with a 3D UNet-like [67] backbone and uses sparse convolutions [68] for feature extraction. DKNet localizes instance centroids based on a candidate mining branch and encodes each instance's information into an instance kernel $k_i \in \mathbb{R}^d$. These instance kernels $\{k_i\}_{i=1}^N$ are used as the weights of a few convolution layers to obtain the final instance masks.

**Tracking**     After frame-level panoptic segmentation, we link each frame via using UniTrack [69] for tracking to obtain the final tracked video cubes for each clip for either modality input. Specifically, instead of incorporating an additional appearance model for tracking embedding extraction, we directly utilize the instance kernels $\{k_i\}_{i=1}^{N}$ from the segmentation step of DKNet, or object query features $\{q_i\}_{i=1}^{N}$ from Mask2Former as the tracking embeddings for the association. We find that the instance kernels exhibit sufficient distinctiveness for tracking purposes, even when dealing with different objects belonging to the same semantic class. This is primarily because each instance kernel is designed to maximize the response for a specific instance while suppressing the responses of all other instances, including those with the same semantic class. For a video sequence with the length $T$, the obtained 4D feature tubes are noted as $Q_i = \{q_i^t\}_{t=1}^{T}$.

## 5.2 Relation Modeling: 4D Scene Graph Generation

The object query tubes $Q_i$ and mask tubes $\mathbf{m}_i$ form a bridge between the first and second stages. These feature tubes first pass through a spatial-temporal transformer encoder, which augments them with both global context information from the overall image and global temporal space.

**Spatial-Temporal Transformer Encoder**     To infuse the feature tubes with additional temporal dimension information and characteristics from other objects in the scene, we draw inspiration from the Spatial-Temporal Transformer [70]. A spatial encoder is initially employed. For all objects co-occurring at the same time $t$, a two-layer transformer encoder is applied to the input, comprising all object features specific to time frame $t$. The spatially-encoded feature tube updates the object feature tube into $\{\tilde{q}_i^t\}_{i=1}^{N}$. Subsequently, a temporal transformer encoder updates each object feature tube along the temporal dimension $T$. By leveraging both the spatial and temporal encoders, we obtain the final feature tube $\{\hat{q}_i^t\}_{i=1}^{N}$, ready for relation training.

**Relation Classification Training**     To train the relation model based on the updated query tube, a training set for relation training must be constructed. It is worth noting that the relation annotation in the training set is in the form of "object-1 relation object-2", with the mask tube of both objects annotated. To start, we associate the updated query tube with ground truth objects. For each ground truth tube, we find the most suitable updated query tube by calculating the video Intersection over Union (vIOU) between ground truth mask tubes, and assign the query feature tube to the respective objects. A frame-level predicate classification is conducted with the assistance of a lightweight fully-connected layer. The inference of the relation classification component simply computes the relation probability between pairs of $\hat{q}_i^t$ and $\hat{q}_j^t$.

# 6   Experiments

Table 2 presents the results of experiments conducted on the PSG-4D dataset. For RGB-D sequences, an ImageNet-pretrained ResNet-101 serves as both the RGB and depth encoder. We set the training duration to 12 epochs. The DKNet, trained from scratch, requires a longer training period of 200 epochs. In the second stage, both spatial and temporal transformer encoders span two layers, and training continues for an additional 100 epochs. Besides the standard PSG4DFormer, we also examine variants with the temporal encoder removed (denoted as "/t") and the depth branch removed (denoted as "/d"). As a baseline, we use the 3DSGG model [18], which employs a GNN model to encode frame-level object and relation information, without considering temporal data.

**RGB-D *vs*. Point Cloud Input**     Table 2 is divided into two sections. The upper part (#1-#3) reports results from point cloud input, while the latter part (#4-#7) details results from the RGB-D sequence. It appears that the RGB-D sequence generally yields better results than the point cloud sequence, particularly for the PSG4D-GTA dataset. This could potentially be attributed to the ResNet-101 backbone used for the RGB-D data, which being pretrained on ImageNet, exhibits robust performance on complex datasets like PSG4D-GTA. Meanwhile, the PSG4D-HOI dataset seems to offer a more consistent scenario with abundant training data, thus narrowing the performance gap between the point cloud and RGB-D methods.

**Significance of Depth**     The results in Table 2 also allow us to evaluate the importance of depth in the RGB-D method. Specifically, we designed a variant of PSG4DFormer (marked as "/d") that

Table 2: **Main Results on PSG4D.** Experimental results are reported on both the PSG4D-GTA and PSG4D-HOI datasets. In addition to comparing with traditional 3DSGG methods, we conduct experiments to compare the PSG4DFormer and its variants. This includes a version with the temporal encoder removed (denoted as "/t") and one with the depth branch removed (denoted as "/d").

| Input Type | Method | PSG4D-GTA | | | PSG4D-HOI | | |
|---|---|---|---|---|---|---|---|
| | | R/mR@20 | R/mR@50 | R/mR@100 | R/mR@20 | R/mR@50 | R/mR@100 |
| Point Cloud Sequence | #1 3DSGG [18] | 1.48 / 0.73 | 2.16 / 0.79 | 2.92 / 0.85 | 3.46 / 2.19 | 3.15 / 2.47 | 4.96 / 2.84 |
| | #2 PSG4DFormer$^{/t}$ | 2.25 / 1.03 | 2.67 / 1.72 | 3.14 / 2.05 | 3.26 / 2.04 | 3.16 / 2.35 | 4.18 / 2.64 |
| | #3 PSG4DFormer | 4.33 / 2.10 | 4.83 / 2.93 | 5.22 / 3.13 | 5.36 / 3.10 | 5.61 / 3.95 | 6.76 / 4.17 |
| RGB-D Sequence | #4 3DSGG [18] | 2.29 / 0.92 | 2.46 / 1.01 | 3.81 / 1.45 | 4.23 / 2.19 | 4.47 / 2.31 | 4.86 / 2.41 |
| | #5 PSG4DFormer$^{/t}$ | 4.43 / 1.34 | 4.89 / 2.42 | 5.26 / 2.83 | 4.44 / 2.37 | 4.83 / 2.43 | 5.21 / 2.84 |
| | #6 PSG4DFormer$^{/d}$ | 4.40 / 1.42 | 4.91 / 1.93 | 5.49 / 2.27 | 5.49 / 3.42 | 5.97 / 3.92 | 6.43 / 4.21 |
| | #7 PSG4DFormer | 6.68 / 3.31 | 7.17 / 3.85 | 7.22 / 4.02 | 5.62 / 3.65 | 6.16 / 4.16 | 6.28 / 4.97 |

doesn't utilize the depth branch. In other words, both the RGB encoder and the SA-Gate are removed, turning the pipeline into a video scene graph generation pipeline. The performance of this variant is inferior compared to the original, which highlights the significance of depth information in the scene graph generation task.

**Necessity of Temporal Attention**     Table 2 includes two methods that do not utilize temporal attention. Specifically, the 3DSGG baseline learns interactions between static object features using a graph convolutional network, while PSG4DFormer$^{/t}$ removes the temporal transformer encoders. The results demonstrate that ignoring the temporal component could lead to sub-optimal outcomes, emphasizing the importance of temporal attention in 4D scene graph generation.

# 7   Real-World Application

This section illustrates the deployment of the PSG-4D model in a real-world application, specifically within a service robot. It extends beyond theoretical concepts and computational models, delving into the practical integration and execution of this cutting-edge technology. As shown in Figure 4, the focus here is to demonstrate how the robot leverages the PSG-4D model (pretrained from PSG4D-HOI, RGB-D input) to interpret and respond to its surroundings effectively.

**Interaction with Large Language Models**     The recent advancements in large language models (LLMs) have displayed their exceptional capabilities in reasoning and planning [71]. LLMs have been utilized as planners in numerous recent studies to bridge different modalities, paving the way for more intuitive and efficient human-machine interaction [72]. In this work, we employ GPT-4 [71], as the primary planner. Designed to align with human instruction, GPT-4 communicates with the robot by translating the raw scene graph representations into comprehensible human language. Therefore, the interaction begins with the prompt, "I am a service robot. For every 30 seconds, I will give you what I have seen in the last 30 seconds. Please suggest me what I could serve." Subsequently, every 30 seconds, the robot engages with GPT-4, providing an update: "In the past 30s, what I captured is: <from start_time to end_time, object-1 relation object-2>, <...>, <...>." This enables GPT-4 to analyze the situation and provide appropriate feedback.

**Post-Processing for Execution**     The effective deployment of the PSG-4D model necessitates a robust set of predefined actions that the robot can execute. Currently, the action list includes tasks such as picking up litter and engaging in conversation with individuals. After GPT-4 provides its suggestions, it is further prompted to select a suitable action from this predefined list for the robot to execute. However, the flexibility of this system allows for the expansion of this action list, paving the way for more complex and varied tasks in the future. To encourage community involvement and the development of fascinating applications, we also release the robot deployment module alongside the PSG4D codebase. The demo robot is priced at approximately $1.2K and comes equipped with an RGB-D sensor, microphone, speakers, and a robotic arm.

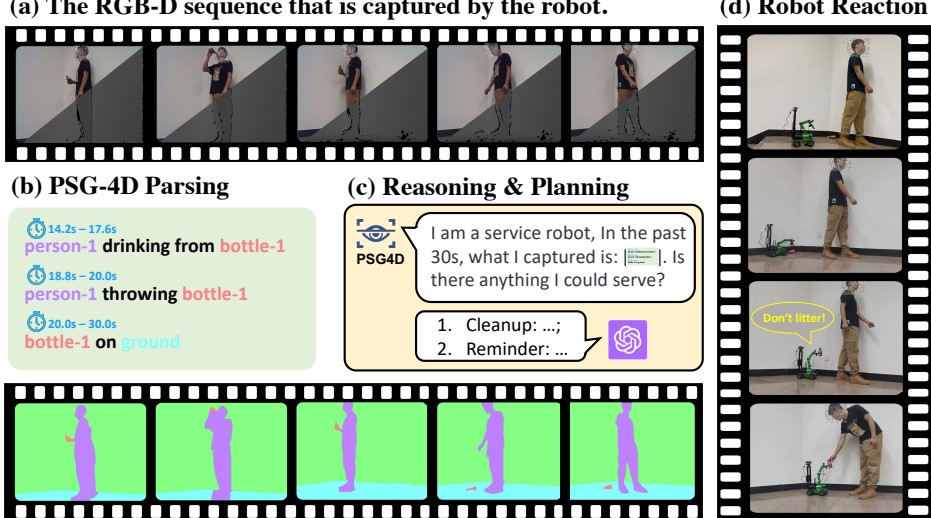

Figure 4: **Demonstration of a Robot Deployed with the PSG-4D Model.** The service robot interprets the RGB-D sequence shown in (a), where a man is seen drinking coffee and subsequently dropping the empty bottle on the ground. The robot processes this sequence, translating it into a 4D scene graph depicted in (b). This graph comprises a set of temporally stamped triplets, with each object associated with a panoptic mask, accurately grounding it in 3D space. The robot regularly updates its PSG4D to GPT-4, awaiting feedback and instructions. In this scenario, GPT-4 advises the robot to clean up the discarded bottle and remind the man about his action. This directive is translated into robot action, as visualized in (d).

# 8 Conclusion, Challenges, and Outlook

This paper presents a novel and demanding extension to the traditional scene graph generation, the 4D Panoptic Scene Graph Generation, which incorporates the spatio-temporal domain into the framework. We introduce a comprehensive framework, the PSG4DFormer, capable of processing both RGB-D and point cloud sequences. The successful deployment of this pipeline in a practical service robot scenario underscores its potential in real-world applications. However, these achievements also highlight the nascent state of this field, emphasizing the necessity for continued advancements to fully exploit the potential of 4D Panoptic Scene Graph Generation.

**Challenges** Despite encouraging results, we have also revealed several persistent challenges in the realm of 4D Panoptic Scene Graph Generation. Through our demonstration, we found that current models, whether derived from PSG4D-GTA or PSG4D-HOI, can handle only simple scenes and falter when faced with more complex real-world environments. Notably, there exist robust models trained in the 2D world. Finding effective and efficient strategies to adapt these models to the 4D domain presents a compelling direction for future exploration.

**Outlook** Future work in this field presents several intriguing trajectories. There is a pressing need for more efficient algorithms for 4D Panoptic Scene Graph Generation, which can handle larger and more diverse environments. Equally important is the creation of comprehensive and diverse datasets that would allow more rigorous evaluation and foster advancements in model development. Particularly noteworthy is a recent Digital Twin dataset [73], which promises a high level of accuracy and photorealism, aligning seamlessly with the objectives of PSG4D. This dataset will be incorporated as the third subset of the PSG4D dataset, readily accessible from our codebase. In addition to robotics, as demonstrated by the practical application of PSG4DFormer, we are also exploring its potential as an autonomous player in the GTA game. Actually, our recent endeavor Octopus [58] strives to complete GTA missions by employing a visual-language programmer to generate executable action code. In contrast to the previously passive task completion, the application in this paper actively perceives and understands the environment, showcasing a shift towards autonomy in robotics. Furthermore, Octopus [58] utilizes a 4D scene graph structure to capture environmental information during the visual-language programmer training, exemplifying a practical application of the PSG4D modality.

We eagerly anticipate the future progress in the field of 4D Panoptic Scene Graph Generation and its potential to revolutionize our understanding of real-world dynamics.

**Potential Negative Societal Impacts**   This work releases a dataset containing human behaviors, posing possible gender and social biases inherently from data. Potential users are encouraged to consider the risks of oversighting ethical issues in imbalanced data, especially in underrepresented minority classes. Nevertheless, all the NSFW content is removed from the dataset.

## Acknowledgement

This study is supported by the Ministry of Education, Singapore, under its MOE AcRF Tier 2 (MOE-T2EP20221-0012), NTU NAP, and under the RIE2020 Industry Alignment Fund – Industry Collaboration Projects (IAF-ICP) Funding Initiative, the National Key R&D Program of China under grant number 2022ZD0161501, as well as cash and in-kind contribution from the industry partner(s).

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
