# OpenReview forum: "4D Panoptic Scene Graph Generation"
_NeurIPS.cc/2023/Conference — NeurIPS 2023 spotlight_

### Official Review · Reviewer_4B6f · 2023-07-05

**Soundness:** 3 good
**Presentation:** 4 excellent
**Contribution:** 3 good
**Rating:** 7
**Confidence:** 4

**Summary:**

This paper offers a novel perspective on understanding a 4D environment via the introduction of 4D Panoptic Scene Graph (PSG-4D). PSG-4D transforms 4D sensory data into nodes, representing entities with precise location and status data, and edges that capture temporal relationships. The authors have developed a high-quality PSG-4D dataset, which includes 3K RGB-D videos with a total of one million frames. Each frame comes with 4D panoptic segmentation masks and detailed, dynamic scene graphs. In order to solve PSG-4D, they propose PSG4DFormer, a model capable of predicting panoptic segmentation masks, tracking masks across the time axis, and generating corresponding scene graphs via a relation component. Extensive experiments demonstrate that their method forms a robust baseline for PSG-4D. Furthermore, the paper provides a real-world application example demonstrating how dynamic scene understanding can be achieved by incorporating a large language model into the PSG-4D system.

**Strengths:**

1. This paper is significant for several reasons: it formulates a new problem, annotates a novel dataset, and provides simple baselines. These aspects could advance future research and encourage applications in robotic systems. Even though the research appears to be a logical progression from panoptic scene graph generation and panoptic video scene graph generation, the introduction of a new data modality provides potential for a broader range of applications. SGG in 4D space is an area that has not been extensively explored, to my knowledge.
2. The dataset selection and construction pipeline is detailed and appears to be reasonable.
3. The paper is well articulated and easy to comprehend. The incorporation of the large language model (LLM) in the proposed application seems to be an intriguing and viable addition.

**Weaknesses:**

1. I have a significant concern with the Experimental section of the paper. The task of Scene Graph Generation is intricate, particularly when extended to a 4D scenario. In building a pipeline to solve this problem, it is reasonable to use various components each serving a unique role. However, it remains unclear which component holds the most promise for future research focus. While the ablation study in Table 2 gives us insights into the depth map data and naïve temporal modeling, it would be valuable for the authors to provide more detailed analysis and insights.
2. In Line 112, there is a minor formatting error: 'M' should be formatted as an equation rather than plain text.

**Questions:**

Please refer to the weaknesses.

**Limitations:**

The authors mentioned potential challenges and societal impact.

---

> ### Author Response · Authors · 2023-08-19
> **Additional Analysis**
>
> We sincerely thank the reviewers for their detailed feedback and recognition of our paper's potential impact. We appreciate the positive remarks regarding the significance, novelty, and presentation of our work. Below, we address your concerns:
>
> **1. Detailed Analysis in Experimental Section:**
>
> Over the past weeks, we've enriched our ablation studies. While Table 2 highlights:
>
> - A superior performance with RGB-D input over Point Cloud input.
> - The beneficial influence of the additional depth channel compared to RGB input alone.
> - The critical role of temporal attention in our relation model.
>
> We aim to provide a more extensive analysis on the PSG4D-GTA dataset encompassing:
>
> - **Correlation between Segmentation, Tracking, and SG Performance**:
>
>     The subsequent table presents intermediate results, showcasing frame-level segmentation results (IOU) and video-level segmentation results that incorporate tracking (vIoU).
>
>     - IoU evaluates all frames individually using a traditional semantic segmentation protocol.
>     - vIoU assesses video segmentation performance, calculating the highest volume IOU for a GT mask tube.
>
>     Our findings indicate that:
>
>     - Improved image segmentation tends to enhance vIoU and SG performance.
>     - The relation recall performance gap is narrower than that of video segmentation. For instance, while Mask2Former significantly boosts video segmentation performance, the subsequent relation modeling stage reduces this gain. This emphasizes the need for improved relation modeling strategies.
>     - We also note that vIoU might not adequately support relation modeling, highlighting a need for community-driven advancements in video segmentation performance.
>     - Considering the limitations of current methods, we contemplate organizing competitions to promote this new task and establish benchmarks.
>
>     | Method | Input | IoU | vIoU | R/mR@20 |
>     | --- | --- | --- | --- | --- |
>     | DKNet | PC | 54.77 | 27.30 | 4.33 / 2.10 |
>     | Mask2Former | RGB | 62.08 | 30.72 | 4.40 / 1.42 |
>     | Mask2Former | RGBD | 70.23 | 37.22 | 6.68 / 3.31 |
>
> - **Class-level Segmentation Analysis**:
>     - Our results display that the DKNet, reliant solely on point clouds, struggles with mean IoU performance, primarily due to challenges in segmenting diverse objects.
>     - Performance disparities are evident for small objects like cups and chairs when solely using point clouds, while larger or structured objects such as persons and potted plants exhibit better segmentation.
>
>     | Method | Input | mIoU | person | car | potted plant | bottle | cup | chair | door |
>     | --- | --- | --- | --- | --- | --- | --- | --- | --- | --- |
>     | DKNet | PC | 13.68 | 93.41 | 51.06 | 80.67 | 5.28 | 1.95 | 22.29 | 2.31 |
>     | Mask2Former | RGB | 40.22 | 95.24 | 57.73 | 77.90 | 43.91 | 45.36 | 59.17 | 52.01 |
>     | Mask2Former | RGBD | 45.14 | 96.39 | 58.45 | 83.21 | 48.35 | 51.04 | 62.23 | 59.14 |
>
> - **Class-level Relation Analysis**:
>     - Our analysis indicates that while relations like **`talking to`** and **`looking at`** are effectively recognized since they involve persons (easily segmented objects), challenges arise for relations like **`holding`** and **`opening`**, especially with the DKNet. This pinpoints a need for strategies addressing relations involving smaller objects.
>     - Also, the performance of `talking to` and `looking at` is also far from saturated.
>
>     | Method | Input | talking to | looking at | holding | opening |
>     | --- | --- | --- | --- | --- | --- |
>     | DKNet | PC | 14.67 | 7.68 | 0.34 | 0 |
>     | Mask2Former | RGB | 18.59 | 9.47 | 2.31 | 0 |
>     | Mask2Former | RGBD | 19.24 | 12.03 | 2.31 | 0 |
>
>
>
> We trust these in-depth analyses will illuminate promising directions for subsequent research building on our foundation.
>
> **2. Formatting Error on Line 112:**
>
> We sincerely apologize for this oversight and thank you for bringing it to our attention. The formatting error will be rectified in the final manuscript.
>
> Once again, we're grateful for your constructive feedback and the time invested in reviewing our work.

---

> ### Comment · Area_Chair_YhZH · 2023-08-21
> **Can you please check the rebuttal comments?**
>
> Dear reviewer,
>
> The authors have provided a response to your comments. Can you please take a look and accordingly comment, and updated your review?
>
> Thanks,
> -Area Chair

---

### Official Review · Reviewer_z3XZ · 2023-07-05

**Soundness:** 3 good
**Presentation:** 4 excellent
**Contribution:** 3 good
**Rating:** 5
**Confidence:** 4

**Summary:**

This paper introduces the concept of the 4D Panoptic Scene Graph (4D-PSG) as an approach for achieving a high-level visual understanding of 4D scenes. Additionally, the authors have developed two datasets annotated with 4D-PSG information. Furthermore, the authors propose a PSG4DFormer method for constructing the 4D-PSG from video data. The utilization of the 4D-PSG has the potential to enhance a robot's ability to comprehend its environment and appropriately respond to it.


**Strengths:**

1. This paper is well written and easy to follow.
2. This paper contributes to the open source community they would release dataset, codebase and robot module.

**Weaknesses:**

1. The novelty of this work is limited. The 4D-PSG is a simple extension from the existing 3DSG. The proposed PSG4DFormer is also a simple combination of the 4D Panoptic Segmentation Model and the Relation Model.
2. Although the authors mentioned the potential application of the 4D-PSG, no experiments are provided to substantiate this claim. The authors really need to add these experiments. Also, a strong baseline should be provided for comparison.


**Questions:**

None


**Limitations:**

The authors have discussed the societal impact.

---

> ### Author Rebuttal · Authors · 2023-08-10
>
> Thank you for your thoughtful feedback. We sincerely appreciate the time you have taken to review our work and provide constructive insights.
>
> **1. Novelty Concerns:**
> We understand your concerns about the perceived linear progression from 3DSG to our proposed PSG4D. However, it's crucial to elucidate that the evolution from 3DSG to PSG4D is not just a dimensional increment but a paradigm shift in scene understanding. While 3DSG provides a hierarchical representation of static three-dimensional environments, PSG4D delves into dynamic realms, capturing the fluidity of real-world interactions over time.
>
> For instance:
>
> - **Dynamic Interactions:** While 3DSG may depict a person next to a bicycle, PSG4D can elucidate sequences such as a person parking a bicycle, then tying their shoelaces, and eventually walking away. Such representations are pivotal for systems requiring predictive or reactive capabilities.
> - **Temporal Relations:** Objects and entities don't exist in isolation. They interact, and these interactions vary over time. Consider two people meeting; 3DSG would represent them standing together, but PSG4D can provide a timeline: Person A walking towards Person B, both shaking hands, and then engaging in a conversation. Such granularity is crucial for applications in surveillance, human-robot interaction, or even cinematic content generation.
> - **Real-world Ambiguities:** Dynamic scenes introduce challenges like occlusion, rapid motion, and interaction complexities. PSG4D not only captures entities but also traces these entities through the ambiguities, ensuring consistent scene understanding.
>
> The transition to a 4D dynamic world from a static 3D realm indeed paves the way for numerous new opportunities and tasks, making this extension substantive and not merely incremental.
>
> **2. Experiments and Baselines:**
> While we recognize the need for further experimental demonstrations, it's essential to highlight our existing work with baselines and experimental variations.
>
> - **3DSGG as a Baseline:** We have indeed incorporated 3DSGG as a fundamental baseline. Specifically, 3DSGG leverages the same panoptic segmentation as our model but diverges in the second stage. Instead of the temporal aspect, it uses a Graph Neural Network (GNN) to encode frame-level object and relation information, inherently missing out on the dynamic temporal nuances.
> - **Model Variations for Analysis:** As elucidated in our discussions (refer Table 2), we did not merely present PSG4DFormer in isolation. We included variations of our model, such as PSG4DFormer/t, which removes the temporal transformer encoders. The comparative outcomes with these models underscore the pivotal role of temporal attention in 4D scene graph generation.
> - **Future Work:** We envision our provided baselines of 3DSGG and the newly introduced PSG4DFormer as foundational building blocks. We are optimistic that they will catalyze further research, fostering advancements in scene graph generation that leverage both spatial and temporal dimensions.
>
> We genuinely hope these clarifications underline our commitment to rigorous experimentation and our intention to advance the field both through our current contributions and by laying the groundwork for subsequent research.
>
> Thank you once again for your consideration and valuable insights.

---

> ### Comment · Area_Chair_YhZH · 2023-08-21
> **Can you please check the rebuttal comments?**
>
> Dear reviewer,
>
> The authors have provided a response to your comments. Can you please take a look and accordingly comment, and updated your review?
>
> Thanks,
> -Area Chair

---

### Official Review · Reviewer_CrvQ · 2023-07-07

**Soundness:** 3 good
**Presentation:** 4 excellent
**Contribution:** 3 good
**Rating:** 5
**Confidence:** 2

**Summary:**

The author introduces a new task called the 4D Panoptic Scene Graph (PSG-4D), which focuses on predicting 4D panoptic scene graphs from RGB-D video sequences. To support this task, the author provides a unique PSG-4D dataset, encompassing diverse viewpoints through two subsets: a third-view synthetic subset (PSG4D-GTA) and an egocentric real-world subset (PSG4D-HOI), each containing detailed frame labels To address the PSG-4D task, the author proposes a unified two-stage model, the PSG4DFormer, which comprises a feature extractor and a relation learner.

**Strengths:**

- The paper is generally well written and easy to follow.
- The new task/datasets seem novel, and well supported by the proposed PSG4DFormer framework.

**Weaknesses:**

- Less relevant to NeurIPS. Perhaps venues like CVPR/ICCV/SIGGRAPH would be a better fit?
- While I recognize the dataset is one the major contribution here, the novelty of this work is not entirely clearly to me.
- Justification for the choice of dataset composition described in Section 4 seems insufficient. Why we choose this specific composition of dataset beyond convenience? Do we know model trained/evaluated on this dataset can generalize nicely to other use cases (or a specific vertical of applications?)
- The experiments seem to be rather limited as only one baseline, though recent and relevant, is evaluated on this very proposed dataset. Are there smaller, ad-hoc dataset that can be used to evaluate the proposed framework to support its external validity?

**Questions:**

See those raised in Weaknesses.

**Limitations:**

Yes, though I’d still would like to hear from the author about the generalizability (even for a specific domain) and external validity of the proposed dataset and framework.

---

> ### Author Rebuttal · Authors · 2023-08-10
>
> #
>
> Thank you for your constructive feedback. We deeply appreciate the time and effort you invested in reviewing our work, and we would like to address each of your raised concerns.
>
> **1. Relevance to NeurIPS:**
> While we understand the viewpoint on the relevance of our work to specific venues, we believe our introduction of a novel task, dataset, and the accompanying framework aligns well with NeurIPS’ emphasis on innovative methods and broader impacts. The 4D application of scene graphs, in essence, tackles a pressing deep learning problem with implications spanning a wide array of AI domains.
>
> **2. Clarity of Novelty:**
> We would like to elucidate that our proposed PSG4D is not merely an enhancement of existing scene graphs. While 3D scene graphs encapsulate the spatial dimension, PSG4D captures the dynamic 4D world, unearthing new opportunities and challenges. This innovative shift is pivotal as it sets the stage for potential future works, especially in the burgeoning field of embodied AI.
>
> For instance, PSG4D-GTA is designed with the vision of enhancing game AI agents. Such agents, equipped with the capability to perceive and comprehend their in-game surroundings holistically, can perform actions with a level of sophistication previously unattained. Similarly, PSG4D-HOI was tailored to be compatible with emerging technologies like egocentric smart glasses (e.g., Apple Vision Pro). By offering 4D perception, the AI can assist users in dynamic, real-world environments with unprecedented accuracy. Our ongoing efforts to incorporate the **[Digital Twin](https://www.projectaria.com/datasets/adt/)** dataset resonate with the same vision.
>
> **3. Dataset Composition Justification:**
> Our selection was not predicated solely on convenience. The synthetic subset, PSG4D-GTA, with its richness in scenes, ensures robust training and sets a strong foundation for applications in game AI, as mentioned. On the other hand, PSG4D-HOI, capturing naturalistic human-object interactions, becomes a cornerstone for real-world applications like smart glasses. Their combined strengths balance controlled and realistic scenarios, aiming at a wide applicability spectrum.
>
> **4. Experiments:**
>
> We sincerely appreciate your suggestion regarding external validation on other datasets. Nonetheless, we wish to emphasize the inherent challenges tied to finding apt scene graph datasets that would align with the novel nature of PSG4DFormer. Most traditional 3D scene graph datasets primarily focus on static settings, devoid of dynamic objects, which essentially contrasts what PSG4DFormer is specifically designed for.
>
> We concede that verifying our methodology across various datasets would indeed strengthen our assertions. However, our primary intention with this submission is to pioneer a new task, furnish the community with robust datasets and benchmarks, and set the groundwork for future research. The novelty and ambition of our work meant we ventured into territories with limited available data that fit the exact criteria our model was built for.
>
> In recognizing the importance of varied datasets for holistic model validation, we're currently making strides in incorporating **[Digital Twin](https://www.projectaria.com/datasets/adt/)**, hopeful that its inclusion will bolster the validity of all PSG4D methodologies.
>
> Our aspiration is that our introduction will spur the community's innovation in areas like robot perception and game/embodied AI development, as elaborated earlier. As the field matures, we are optimistic about more comprehensive benchmarks and comparative studies.

---

> > ### Comment · Reviewer_CrvQ · 2023-08-15
> >
> > I thank the author for their response. However, without seeing the claimed addition (e.g., the Digital Twin dataset) to this dataset, I still find the work having room for improvement in terms of data source diversity and a more comprehensive evaluation on the experimental side. I'll keep my score as is.
> >
> > That said, given the overall positive reviews of other reviewers, I believe this paper has a fair chance of getting accepted, which despite my reservation above, would still be a valuable contribution to the community.

---

> > > ### Author Response · Authors · 2023-08-19
> > > **Thank you for your feedback!**
> > >
> > > Thank you for your feedback and for taking the time to review our paper. We genuinely appreciate your insights and the constructive nature of your remarks. We are also happy to update our added experimental analysis within the [comment to Reviewer 4B6f](https://openreview.net/forum?id=GRHZiTbDDI&noteId=NZP8e51cJt).
> > >
> > > Regarding the data source diversity concern, we are committed to enhancing our work in this direction. Apart from the addition of the Digital Twin dataset, would you be kind enough to provide any specific suggestions or pointers that you believe would address this aspect further? We believe your insights will be instrumental in refining our approach and expanding the scope of our evaluation.

---

### Official Review · Reviewer_2QCE · 2023-07-07

**Soundness:** 3 good
**Presentation:** 3 good
**Contribution:** 4 excellent
**Rating:** 8
**Confidence:** 4

**Summary:**

This paper introduces a new task of scene graph generation that predicts 4D panoptic scene graphs from RGB-D video sequences. This paper also provides a high-quality PSG-4D dataset including PSG4D-GTA with third-view synthetic data and PSG4D-HOI with egocentric real-world data. As the third contribution, this paper also proposes a unified two-stage model to showcase real-world scenarios.

**Strengths:**

1. The new task introduced by this paper is very important. It will help to enable embodied AI to better understand the 4D dynamic real world with complex object-object relations.

2. As illustrated in Tab. 1, the collected dataset features dynamic scenes with much richer data type than previous datasets, covering both egocentric and third-view data. If released, this new dataset will greatly benefit the community.

3. In section 7, the paper also illustrates how to deploy the PSG-4D model in a real-world application by interacting with large language models. This shows the potential of exploiting the introduced dataset to solve practical problems in the real world.

**Weaknesses:**

1. The introduced datasets are based on existing datasets (ie, SAIL-VOS 3D and HOI4D datasets). Therefore, they only cover  synthetic data for third-person views. This may cause the imbalanced development of algorithms for egocentric and third-view data.

2. L187-190 briefly mentions how the relation annotations are first automatically prepared and then reviewed and revised. Could the authors elaborate and share a bit more whether more design choices have been made to make the process more efficient or this is a very tedious/time-consuming task (if so, what is the average/total time needed for each scene or all scenes)?

**Questions:**

It may be a bit easier for readers to digest the table if the numbers from the best-performing model can be highlighted.

**Limitations:**

The authors have adequately addressed the limitations and broader impacts.

---

> ### Author Rebuttal · Authors · 2023-08-10
>
> We genuinely appreciate your comprehensive feedback on our paper, "4D Panoptic Scene Graph Generation." Based on your comments, we'd like to further clarify our intentions and the processes involved:
>
> **Regarding Weaknesses:**
>
> 1. **Datasets based on existing datasets**:
>     - The motivation behind the development of PSG4D is to set the stage for the potential evolution of comprehensive robot perception or embodied AI. In these future scenarios, the agent, whether operating within an egocentric view in the real world or as a game agent in a virtual environment, would need to perceive and understand its 4D surroundings accurately. Given this goal, the emphasis of PSG4D is not necessarily to ensure a balanced representation of both egocentric and third-person view data in our current work. Instead, it is geared towards shaping the potential next steps of agent and robot vision, where an egocentric view becomes highly pertinent. This approach supports the incorporation of datasets like **[Digital Twin](https://www.projectaria.com/datasets/adt/)**, which is rich in egocentric data.
> 2. **Relation Annotations Process**:
>     - Indeed, the relation annotation process is a painstaking endeavor. To streamline and improve the annotation efficiency for SAIL-VOS 3D, we designed a method where annotators first describe the video in a few sentences. This allows them to grasp the essence and primary interactions in the content. By then annotating relations based on these descriptions, we can ensure that the main interactions and relations are captured, while minimizing trivial positional relations that might not be as informative.
>     - The manual nature of this task requires careful attention, especially when determining the start and end frames for each relation. On average, a 60-second video from SAIL-VOS 3D requires about 30 minutes for annotation. For the HOI4D dataset, given that it already contains annotations (albeit with quality variations and missing object-object relations), our annotators focused on refining and supplementing these existing annotations. Typically, a 20-second video from HOI4D takes about 5 minutes to revise.
>
> **Addressing Questions**:
>
> - **Highlighting best-performing model in tables**: We are grateful for this suggestion. In our revised version, we will highlight the leading results for better clarity and ease of understanding.
>
> In summary, our overarching aim with this work is to bolster the capabilities of future embodied AI systems, ensuring they can navigate and understand dynamic 4D environments, be it real or virtual. We hope our clarifications address your concerns, and we are enthusiastic about the potential positive impact of our work on the AI community.

---

> > ### Comment · Reviewer_2QCE · 2023-08-17
> >
> > Thanks for the responses! After reading all reviewers' comments and the responses from the authors, I am leaning towards keeping my original rating.

---

### Official Review · Reviewer_PY8p · 2023-07-10

**Soundness:** 3 good
**Presentation:** 3 good
**Contribution:** 4 excellent
**Rating:** 7
**Confidence:** 5

**Summary:**

This paper proposes a new task, named 4D panoptic scene graph generation (PSG-4D). At the same time, a dataset, including PSG4D-GTA and PSG4D-HOI, is collected for this task, and a benchmark is provided to evaluate the performance of this task. PSG-4D offers a more comprehensive scene understanding, bringing more thought and possibilities for the practical implementation of scene understanding. The paper shows the potential application of PSG-4D in Robot Deployed scenarios by combining it with LLM, which is really meaningful.

**Strengths:**

1. A new task, dataset, and benchmark are proposed to solve the PSG-4D problem, which provides a more comprehensive scene understanding in the real world.
2. This paper shows the potential applications of the PSG-4D task, making it appear more practically significant.

**Weaknesses:**

The sample size of the dataset and the number of relationship labels are somewhat limited, and the recall provided by the benchmark is really low, which may limit its practical application.

**Questions:**

Please refer to the weaknesses.

**Limitations:**

Not apply.

---

> ### Author Rebuttal · Authors · 2023-08-10
>
> Thank you for your detailed review and for recognizing the contributions of our work on "4D Panoptic Scene Graph Generation". We are grateful for your feedback and would like to provide additional information to address your concerns.
>
> **Regarding Weaknesses:**
>
> 1. **Sample size of the dataset**:
>     - As a continuous project, we are fully aware of the dataset size limitations and have plans for its expansion. We aim to provide the research community with a growing, high-quality dataset to further the advancements in PSG-4D.
>     - We are currently in the process of incorporating the **[Digital Twin](https://www.projectaria.com/datasets/adt/)** dataset, an egocentric dataset with high-quality panoptic segmentation annotation and RGBD signals, which aligns well with PSG4D requirements. Our vision for the future is to integrate more such high-quality datasets to provide a more comprehensive setting for PSG-4D research.
> 2. **Number of relationship labels**:
>     - The design of our relationship labels was to capture comprehensive visual expressions from the data. Based on our dataset, we believe that the existing relationships can aptly describe most scenarios.
>     - Our method of introducing relations begins with a pre-defined relation set, typically derived from the PSG dataset. During the annotation phase, if annotators identify potential new relations, they coordinate with the primary authors to evaluate and decide on their inclusion. This process ensures that our relation set remains both diverse and relevant.
>     - We recognize the potential for encountering novel interactions or relations when testing beyond our current dataset. This recognition points towards an exciting direction for future research: open-vocabulary relation predictions. We hope this challenge invites the community to explore and innovate further in this domain.
> 3. **Benchmark recall**: Our goal with introducing PSG4DFormer as a baseline was to provide the community with a starting point. While we acknowledge the recall's current limitations, we believe that the evolving nature of our dataset and the advent of new methodologies will lead to improved recall metrics.
>
> We hope that the above elucidation addresses your concerns related to the dataset size, the relationship labels, and the recall performance. We remain open to further discussions and clarifications on any aspect of our work.

---

> > ### Comment · Reviewer_PY8p · 2023-08-18
> >
> > The novel task presents a potential direction for completely understanding scenes with scene graphs. However, I still have concerns about the dataset being potentially challenging to follow due to limited scale. Nonetheless, I maintain my acceptance score.

---

### Decision · Program_Chairs · 2023-09-21

**Decision:**

Accept (spotlight)

**Comment:**

The submission presents a representation, named 4D Panoptic Scene Graph Generation, which extends the 3D scene graph into the temporal domain. 3D scene graphs have been attracting interest as one solution to filling the gap between lower-level information that is often extracted from sensory data, like images, and higher-level information that is needed for performing action. An extension to incorporate the temporal domain is naturally useful. The paper includes a number of relevant components in this area, including a dataset, baselines, and demonstration of applications.

All reviewers are positive. The paper has unanimous acceptance recommendations. The rebuttal was also considered and generally appreciated by the reviews. The less positive reviewers cited issues with novelty and relevance to NeurIPS compared to vision conferences. The AC has considered them and judged the extension to the temporal domain presents significant enough challenges to not view the submission as too incremental.

As the paper poses to be the first to introduce the 4D extension, the main question for the future impact of the paper, and generally the proposed direction, is how far the utility can be pushed and experimentally demonstrated.